# Effects of Harvesting Grabbing Type on Grabbing Force and Leaf Injury of Lettuce

**DOI:** 10.3390/s23136047

**Published:** 2023-06-29

**Authors:** Yidong Ma, Pengzhan Hu, Xinping Li, Xin Jin, Huankun Wang, Chao Zhang

**Affiliations:** College of Agricultural Equipment Engineering, Henan University of Science and Technology, Luoyang 471003, China; 200320041515@stu.haust.edu.cn (P.H.); aaalxp@126.com (X.L.); whk430@outlook.com (H.W.); zchao@haust.edu.cn (C.Z.)

**Keywords:** hydroponic lettuce, harvesting, flexible finger, grabbing force, leaf injury

## Abstract

Hydroponic lettuce is the main cultivated leafy vegetable in plant factories, and its scattered leaves are delicate and easily injured. Harvesting is an important process in the production of hydroponic lettuce. To reduce the injury level of hydroponic lettuce during harvesting, the impacts of the flexible finger-grabbing position applied on the grabbing force and the area of the injured leaves were investigated in this study by utilizing thin-film sensors and a high-speed video camera. According to the overlapping structural characteristics of adjacent leaves on lettuce, flexible finger-grabbing positions were divided into areas of the surface of the leaves and the intersections of the leaves. Three grabbing types—which are referred to in this paper as Grabbing Types A, B, and C—were identified according to the number of flexible fingers grabbing the leaf surface and the intersection area of the leaves. The force curves of all the flexible fingers were measured by thin film sensors, and the injury area of the leaves was detected using an image processing method. The results showed the consistency of the grabbing force curves and the motion characteristic parameters of the four flexible fingers. The maximum grabbing force of each flexible finger appeared at the stage of pulling the lettuce. The grabbing force of the flexible fingers acting on the intersection areas of the leaves was less than that acting on the leaf surface. As the number of flexible fingers acting on the intersection areas of the leaves increased, both the injury area of the leaves and the grabbing force decreased gradually. Grabbing Type C had the smallest injury area of the leaves: 120.3 ± 13.6 mm^2^ with an 11.4% coefficient of variation.

## 1. Introduction

Plant factories are an advanced form of protected horticulture, which have been rapidly developed and widely employed in urban agriculture and other scenes [1,2,3,4,5,6]. Hydroponic lettuce is the most common cultivated leafy vegetables in plant factories, and its leaves are delicate and easily injured [7,8,9,10] Therefore, reducing the degree of injury of hydroponic lettuce is a critical and important task in mechanical harvesting [11,12,13,14,15]. To adapt to the rapid development of plant factories for hydroponic lettuce production, it is important to conduct an investigation to find an optimized low-damage harvesting method.

At present, low-damage harvesting research mainly focuses on the harvesting of fruits and vegetables, and it is primarily achieved through optimizing the harvesting mechanism and employing force sensor technology [16,17,18,19,20,21,22,23]. The flexible mechanism was widely used in the production of fruit picking mechanisms due to its strong adaptability, high conformity, and high flexibility. Rafael et al. designed a mango harvesting device composed of six flexible fingers, based on the criterion of harvesting damage, and its harvesting effect was compared with three rigid mechanisms. The results showed that the harvesting damage to immature mangoes was 0% and to ripe mangoes was 8%, which was better than the rigid mechanism [16]. Wang et al. designed a suction-type flexible mechanical hand for apple picking and conducted experiments using a thin film pressure sensor to study the effect of different suction angles on harvesting efficiency. The results showed that an optimal suction angle could improve the success rate of picking and avoiding apple damage caused by picking failures [17]. Liu et al. designed and fabricated a pineapple harvesting machine with multi-flexible finger rollers, and its flexible components were made of nylon and rubber materials. Field test results showed that its damage rate was 8%, and the rollers made by flexible materials effectively reduced the harvest damage of pineapples [18]. Due to the delicate and easily damaged characteristics of hydroponic lettuce, the design of the harvesting mechanism directly affects the leaves’ damage. According to their structural characteristics, harvesting mechanisms for hydroponic lettuce could be divided into rigid [24,25,26] and flexible one [27]. Gao et al. used a rigid gripping plate to grip the lettuce from the side during harvesting, which caused significant radial deformation of the lettuce and resulted in damage to the lettuce [24]. Ma et al. designed a whole-plant harvesting device for hydroponic lettuce. Two rigid clamping rods were used to gather the leaves of lettuce; they had a good gather effect but easily caused damage to the lettuce leaves at the bottom. Therefore, rigid components had few advantages in low-damage harvesting of lettuce [25]. Ma et al. designed a flexible harvesting device with four flexible fingers. Grabbing lettuce with flexible fingers reduced the harvesting damage compared with rigid mechanisms [27]. The arrangement of hydroponic lettuce leaves has certain characteristics and rules, thus the effects of the lettuce grabbing position on the leaves’ injury is worthy of further research.

To further reduce the injury level of lettuce leaves in flexible harvesting of hydroponic lettuce, the aims of this paper were (1) to sort the grabbing types according to the arrangement of hydroponic lettuce leaves and the flexible fingers’ grabbing positions; (2) to investigate the grabbing force and motion characteristics of flexible fingers in different grabbing types by utilizing thin-film pressure sensors and high-speed cameras; and (3) to analyze how the grabbing positions and the grabbing types affect the injury level of vegetable leaves.

## 2. Materials and Methods

### 2.1. Hydroponic Lettuce for Experiment

The hydroponic lettuce tested in the experiment was the Naiyou lettuce cultivated in a plant factory for 30 days from a corporation, Xutian Photoelectric Co., Ltd., Xi’an, China, as shown in Figure 1. The average height of the hydroponic lettuce plant and the average spread size of the lettuce leaf were 86.5 mm and 195.3 mm, respectively. The leaves were scattered without being knotted. The leaves of the hydroponic lettuce were delicate and prone to injury, and overlap zones existed between the adjacent leaves. To explore the influence on the leaf’s injury brought about by different grabbing positions, the surface of the leaf and the intersection of the areas of the leaves for grabbing were examined in this research.

### 2.2. Experimental Equipment

A flexible harvesting test device for hydroponic lettuce (Figure 2) was mainly composed of a lettuce grabbing mechanism and a stem cutting mechanism. The four pneumatic flexible fingers were the core operating components, evenly arranged at 90-degree angles of the grabbing mechanism. The optimal operational parameters combination of the hydroponic lettuce harvester was a grabbing diametrical ratio of 0.76, a grabbing height ratio of 0.55, and a bending angle of 39.7° [27]. The parameters were adjusted by the grabbing parameters adjustment motor and the input pneumatic pressures of flexible fingers. The grabbing positions were adjusted by the adjustment motor (Figure 2). The impacts of grabbing positions on grabbing forces and the injury of leaves were investigated under the optimal grabbing parameters. 

### 2.3. Experimental Methods

#### 2.3.1. Grabbing Types of Flexible Fingers

Single flexible fingers’ grabbing positions were indicated in Figure 3a, which could be sorted into two scenarios due to the structural characteristics of the overlap zone of adjacent leaves. One was named ‘leaf surface’, and the other was named ‘intersections of leaves’. To research the impact on leaf injury brought about by the grabbing position of the flexible finger, there were five grabbing types in total, according to the number of flexible fingers that grab the intersection of leaves, theoretically. Four flexible fingers in the grabbing device were evenly distributed around the lettuce, twenty hydroponic lettuce plants were statistically analyzed by utilizing the exhaustive method, and only three grabbing types were found in experimental processes (Figure 3b): Type A—all four fingers grabbed the leaf surface; Type B—three fingers grabbed the leaf surface and the other one grabbed intersections of leaves; and Type C—two flexible fingers grabbed the surface of the leaf and two flexible fingers grabbed the intersections of the leaves. 

#### 2.3.2. Grabbing Force Measurement and Grabbing Process Analysis by High-Speed Camera

The grabbing force of flexible fingers on hydroponic lettuce during harvesting was measured by the grabbing force measuring system (Figure 4). It mainly consisted of thin-film pressure sensors (DF9-40, measuring range: 0–20 N, response: 0.2 N, Suzhou Leanstar Electronic Technology Co., Ltd., Suzhou, China), a signal conversion module (SCM-5, Suzhou Leanstar Electronic Technology Co., Ltd., Suzhou, China), a 16-channel data acquisition card (USB-1608GX-2AO, Measurement Computing Corporation, Norton MA, USA), and a computer. In the pre-experiment, five thin-film pressure sensors were attached to single flexible fingers from top to bottom, but the pressure was only detected at the bottom of the flexible finger. Due to the bending of the flexible fingers, the sensors that did not attach to the finger bottom could not tightly fit to the lettuce, except the bottom. To ensure the accuracy of the grabbing force measurement, the force-sensitive areas (circular area of the sensor) of four thin-film pressure sensors were respectively attached to the bottoms of the flexible fingers (the contact point between the flexible finger and the lettuce). In the process of grabbing the lettuce with the flexible fingers, the resistance received by the film pressure sensor was converted into a voltage signal by the signal conversion module. Four measurement signals were collected by the data acquisition card at the same time, and the log of the grabbing force value was recorded in DAQami 4.2.1 software on the computer. When the grabbing force of flexible fingers was measured, tests were carried out according to the grabbing type (i.e., A, B, or C) to measure the grabbing force of the flexible fingers. Experiments were repeated ten times for each grabbing type (one plant of hydroponic lettuce was used in each experiment). A total of thirty plants of hydroponic lettuce were consumed for these experiments.

To analyze the motion characteristics of flexible fingers in the process of grabbing hydroponic lettuce, a PHANTOM high-speed camera (Vision Research Inc., Wayne, NJ, USA) (Figure 5) was utilized to record the grabbing process. The frame rate of the high-speed camera was set to 400 f/s. Figure 6 demonstrates a method to measure the motion characteristic parameters of flexible fingers by utilizing a high-speed photography software (Phantom Camera Control), and the measurement procedure is listed as follows:

(1) Measurement calibration, i.e., the ratio of the actual size of the object to the image pixel (1 mm/pix).

(2) The origin and coordinate system set up, i.e., the center of lettuce planting plate was taken as the coordinate origin and the rectangular coordinate system (*xO*_p_*y*) was established.

(3) Measurement of the motion parameters of the contact point at the end of the flexible finger, i.e., the contact mostly occurred at the end of flexible finger, and, hence, the measurement point was selected. According to the different grabbing parts of the flexible fingers, two measuring points were selected in the test: measuring point (P_1_) was the contact point between the flexible finger end and the intersection area of the leaves, and measuring point (P_2_) was the contact point between the flexible finger end and the cabbage leaf surface. Phantom Camera Control software was used to capture the motion trajectories of both measuring points during the grabbing process (recorded every 0.25 s) and to calculate the displacement, velocity, and acceleration of the measuring points. 

#### 2.3.3. Measurement of Leaves Injury Area by Image Processing

The injured areas of vegetable leaves were measured under various grabbing types (i.e., A, B, and C). Experiments were repeated ten times for each grabbing type, and one plant of hydroponic lettuce was tested in each experiment. A total of thirty plants of hydroponic lettuce were consumed. The injured area was defined as the total area of injured parts of the lettuce leaves of a single plant after harvesting. Since the injury on the upper part of the vegetable leaves was difficult to observe directly, fuchsin staining was used to highlight the injury [25,27]. An image processing tool was used to measure the area of the injured parts of vegetable leaves. The measurement procedure is listed as follows: 

(1) Manually peel leaves off from the harvested lettuce.

(2) Dye the leaves for 15 min in the 5% acid fuchsin.

(3) Rinse the magenta off from surfaces of the leaves and then dry them afterwards.

(4) A Microsoft LifeCam camera was used to collect the images of vegetable leaves, and the height of the camera from the upper surface of the loading board was 350 mm. Nine calibration units with size of 20 × 20 mm were used for area calibration. The calibration results indicated that the average number of pixels in 400 mm^2^ was 3458. The root mean square error and coefficient of variation of pixels were 157 and 4.5%, respectively (Figure 7).

(5) The injured parts of the vegetable leaf in the picture were extracted in Photoshop as the image foreground, and the image background was part of the black area (i.e., the pixels with R, G, and B values of 0). The image was saved as in Figure 8c. According to the area calibration result, the area of image foreground pixels was calculated in MATLAB (i.e., the pixels with R, G, and B values all greater than 0) (Figure 8d), and, therefore, the injured area of the single vegetable leaf could be calculated.

## 3. Results and Discussion

### 3.1. The Curves of Grabbing Force and Grabbing Process Analysis 

The grabbing force curves of flexible fingers and the characteristics of high-speed photography movement were analyzed. Taking Grabbing Type B as an example, Figure 9 shows the measurement results of the four flexible fingers in the DAQAMI software. Table 1 describes the statistical results of the force values of the four flexible fingers. Figure 10 shows the motion trajectories of the contact points and the measurement results of motion parameters captured in the Phantom Camera Control software. Table 2 shows the accelerations of the contact points. The process of grabbing lettuce with flexible fingers can be divided into four stages:

In Stage 1, flexible fingers were approaching the lettuce. At this stage, the flexible fingers began to approach the lettuce in a negative pressure state. Since the flexible finger did not touch the lettuce, the grabbing force was approximately 0 N. Due to the comprehensive effect of flexible finger deformation under negative pressure and the vertical downwards movement of the robotic arm, the maximum speeds of both points P_1_ and P_2_ were about 150 mm/s.

In Stage 2, the flexible fingers grabbed the vegetable leaves. At this stage, the pneumatic pressure in the flexible fingers started to increase, and its value turned from negative to positive. Then, the flexible finger started to contact and grab the lettuce. The average accelerations of P_1_ and P_2_ were 410 and 445 mm/s^2^ with negative and positive directions of the *x* axis, respectively. The grabbing forces of Finger 1 increased rapidly, with the maximum value of 1.72 N, which was smaller than those from Fingers 2, 3, and 4. The resistances of flexible fingers were weakened after contacting the vegetable leaves. The average acceleration of P_1_ and P_2_ dropped to 215 and 220 mm/s^2^ with positive and negative directions of the *x* axis, respectively, and the lettuce state also changed from loose to firm. The average accelerations of both P_1_ and P_2_ were 0 mm/s^2^ when the grabbing process was completed with the positive pneumatic pressure of the fingers. Afterwards, the grabbing forces of flexible Finger 1 and Finger 3 were stabilized at about 1.50 and 1.70 N, respectively. In this stage, the vertical robotic arm remained stationary, and the flexible fingers were the only deformed part. Therefore, the speeds of both contact points P_1_ and P_2_ remained stable. As Table 1 shows, the coefficient of variation (43.9%) of grabbing force for Finger 1 was larger than Finger 2 (41.4%) and Finger 4 (39.6%) and smaller than Finger 3 (55.2%). It indicated that grabbing position had significant influence on the fingers’ grabbing forces.

In Stage 3, the flexible fingers pulled out the lettuce from the board. The flexible fingers grabbed the lettuce and pulled out the lettuce from the hole in the planting board under the action of the vertical mechanical arm. The average accelerations of P_1_ and P_2_ in the process of pulling were 292 and 310 mm/s^2^ with a positive direction of the *y* axis, respectively. The maximum grabbing force of flexible Finger 1 (acting on the intersection area of leaves) was about 2.15 N, which was lower than that of flexible Finger 3 (acting on the surface of the leaf) (about 2.47 N). The grabbing force of the flexible fingers in this stage increased compared with the grabbing force in the second stage due to the occurrence of a vertical friction force between the flexible fingers and the lettuce in the vertical pulling process. The curling process of the flexible finger was completed, and only the vertical manipulator moved upward. The velocities at points P_1_ and P_2_ increased in the first place and then remained stable. As Table 1 and Figure 9 show, the maximal grabbing force (3.23 N) occurred on Finger 2, and the second largest grabbing force (2.79 N) occurred on Finger 4. At this stage, Finger 2 and Finger 4 grabbed the leaves’ surfaces. Fingers needed to overcome the friction resistance of the planting board hole against the root, so the force would fluctuate through the ‘pulled out’ and ‘lifted up’ process. The coefficient of variation of grabbing force for Finger 1 (50.4%) and Finger 3 (59.6%) were larger than that of Finger 2 (41.4%) and Finger 4 (30.4%). Finger 1 and Finger 3 were opposite each other, which indicated that the finger grabbing force would be affected if it was opposite to the finger that was grabbing the intersection area of leaves.

In Stage 4, flexible fingers released the lettuce. At this stage, with the decreasing value of the positive pneumatic pressure of the flexible fingers, the lettuce was released, so the grabbing forces and the speeds of both P_1_ and P_2_ dropped to 0. The above analysis indicates that the grabbing force curves and motion characteristic parameters of the four flexible fingers were basically the same during harvesting. At the same grabbing stage, the grabbing force of the flexible fingers acting on the intersection area of the leaves was smaller than that acting on the leaf surface. The maximum grabbing force appeared in Stage 3. The differences of the grabbing position would cause variation in the maximum grabbing force. In the process of lettuce harvesting, the injury of lettuce leaves is indeed mainly caused by the maximum grabbing force. Therefore, the maximum grabbing force value curve was selected as the data source in the subsequent study, which was about the influence of the grabbing position on the grabbing force.

### 3.2. The Influence of Grabbing Position on Grabbing Force

Figure 11 shows the grabbing force of four flexible fingers of different grabbing types (i.e., A, B, and C). Table 3 shows the average grabbing forces under various grabbing states. For Grabbing Type A, all four flexible fingers acted on the leaf surface, and the average grabbing force was basically the same (about 2.2 N). The standard deviation and coefficient of variation were 0.042 N and 1.9%, respectively. The grabbing force was smaller than that in the study of Wang et al. [28], which used rigid components to grab leaf vegetables, and the average pulling force acting on hydroponic lettuce was 13.03 N. According to the arrangement of lettuce leaves, a reasonable finger grabbing position could reduce the grabbing force. For Grabbing Type B, flexible Finger 1 acted on the intersection areas of the leaves with an average grabbing force (about 1.7 N) that was lower than that of flexible Fingers 2, 3, and 4 (i.e., 2.1, 2.3, and 2.3 N, respectively). The standard deviation and coefficient of variation were 0.302 N and 14.4%, respectively. For Grabbing Type C, the average grabbing forces of flexible fingers 1 and 3 (acting on the intersections of the leaves) were basically the same (about 1.8 N), while the average grabbing forces of flexible fingers 2 and 3 (acting on the leaf surface of the lettuce) were 2.2 and 2.3 N, respectively.

Table 3 shows that among the three grabbing types, flexible fingers 1 and 3 contacted the vegetable leaf surface or the intersection areas of vegetable leaves, and their average grabbing forces were greater than those of flexible fingers 2 and 3, acting on the vegetable leaf surface. The standard deviation and coefficient of variation of the mean grabbing forces of flexible fingers 1 and 3 were also greater than those of flexible fingers 2 and 3. The above results indicated that the grabbing positions of the flexible fingers could affect the grabbing forces. The grabbing forces of flexible fingers acting on intersection areas of the leaves would be smaller than those of the flexible fingers acting on the leaf surface.

As shown in Figure 11, different grabbing positions (leaf surface and intersection areas of leaves) affected the grabbing forces due to the different grabbing area radii of the flexible fingers. The pneumatic pressures of the four flexible fingers were the same when grabbing the lettuce (i.e., 0.06 MPa). The radius of the grabbing areas of the four flexible fingers was the same when there was no load. When grabbing the lettuce, the curling of the flexible fingers with a smaller radius was closer to the no-load state, resulting in a smaller grabbing force. Grabbing Type B was taken as an example for mechanical analysis, as shown in Figure 12. Due to the interleaving arrangement of the leaves, the actual grabbing radius r1 of flexible Finger 1 (acting on the intersection area of the leaves) was smaller than the actual grabbing radius r of flexible fingers 2, 3, and 4 (acting on the leaves), which resulted in the decrease in the interaction force value between flexible Finger 1 and the lettuce. Therefore, the grabbing forces of flexible fingers acting on intersection areas of leaves (F_1_) were smaller than those acting on the surfaces of the leaves (i.e., F_2_, F_3_, F_4_).

### 3.3. Effect of Grabbing Type on Leaves Injury Area

Table 4 shows the injured areas of vegetable leaves under Grabbing Type A, B, and C. For Grabbing Type A, all four flexible fingers acted on the leaves’ surfaces, and the injured areas of leaves were the largest (275.5 ± 16.0 mm^2^), which was smaller than that of Ma et al. [25], who gathered hydroponic lettuce with a rigid gripper rod. For Grabbing Type B, the injured area was 206.8 ± 16.4 mm^2^. Grabbing Type C had the smallest injured area (120.3 ± 13.6 mm^2^), which was smaller than that of Ma et al. [27], who did not distinguish the grabbing positions (186 mm^2^). The most reasonable grabbing type could effectively reduce the grabbing force and the grabbing damage areas of leaves. The injured areas of vegetable leaves of Grabbing Types A, B, and C decreased successively. Analyzing the grabbing forces of flexible fingers in Figure 11, it was found that when the number of flexible fingers acting on the intersection areas of leaves increased, both the grabbing forces of the flexible fingers and the injured areas of the vegetable leaves decreased. The leaf’s damage might be caused by grabbing or bending of fingers. The position of the injured areas shown in Figure 8 was not consistent with the grabbing position, which was because the leaf’s damage was caused by finger bending. According to the research results, the harvesting quality could be improved with the visual identity of grabbing positions, and the grabbing positions could be adjusted by the motor shown in Figure 2.

## 4. Conclusions

(1) According to the arrangement of lettuce leaves, the grabbing position of flexible fingers could be divided into the leaf surface and the intersection of the leaves. There were three grabbing types of the four flexible fingers: all four flexible fingers grab the leaf surface (Type A); three flexible fingers grab the leaf surface and one flexible finger grabs the intersection area of the leaves (Type B); and two flexible fingers grab the leaf surface and two flexible fingers grab the intersection areas of the leaves (Type C).

(2) The grabbing force curves and the motion characteristic parameters (displacement and velocity) of the four flexible fingers were basically the same. At the same grabbing stage, the grabbing forces of the flexible fingers acting on the intersection areas of the leaves were less than the force acting on the leaves’ surfaces. The maximum grabbing force of each flexible finger appeared in the third stage of the grabbing process (flexible finger plucking lettuce).

(3) As the number of flexible fingers acting on the intersection areas of the leaves increased, the grabbing forces of various grabbing types decreased successively and the injured areas of the leaves also decreased accordingly. The injury area of the leaves of Grabbing Type C was the smallest. The mean value of the injury area of Grabbing Type C was 120.3 mm^2^. The standard deviation and coefficient of variation were 13.6 mm^2^ and 11.4%, respectively.

## Figures and Tables

**Figure 1 sensors-23-06047-f001:**
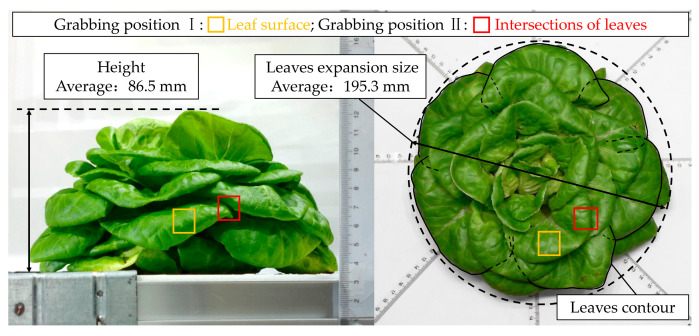
The hydroponic lettuce.

**Figure 2 sensors-23-06047-f002:**
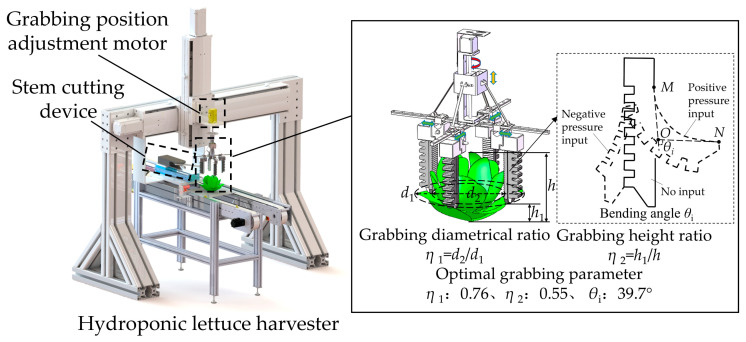
Flexible harvesting test device for hydroponic lettuce.

**Figure 3 sensors-23-06047-f003:**
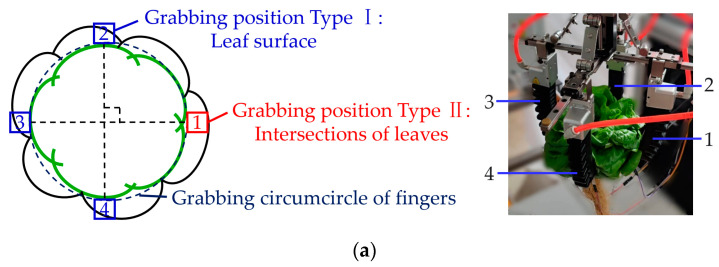
Grabbing positions and types of flexible fingers: (**a**) grabbing positions of a single flexible finger; (**b**) grabbing types of flexible fingers. Notes: 1–4 were pneumatic flexible fingers.

**Figure 4 sensors-23-06047-f004:**
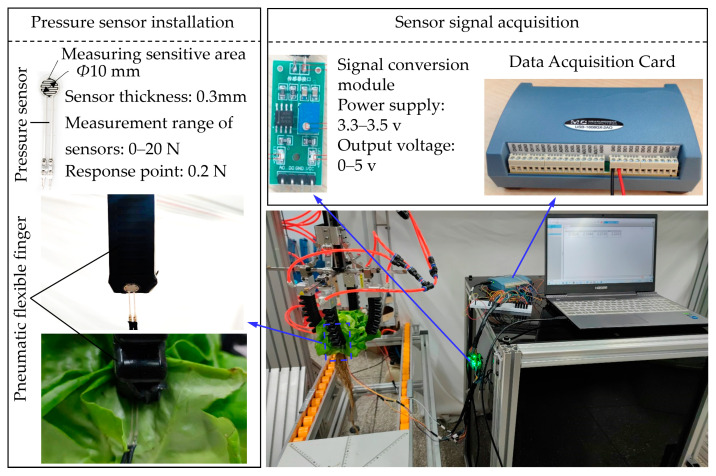
Grabbing force measurement system for flexible fingers.

**Figure 5 sensors-23-06047-f005:**
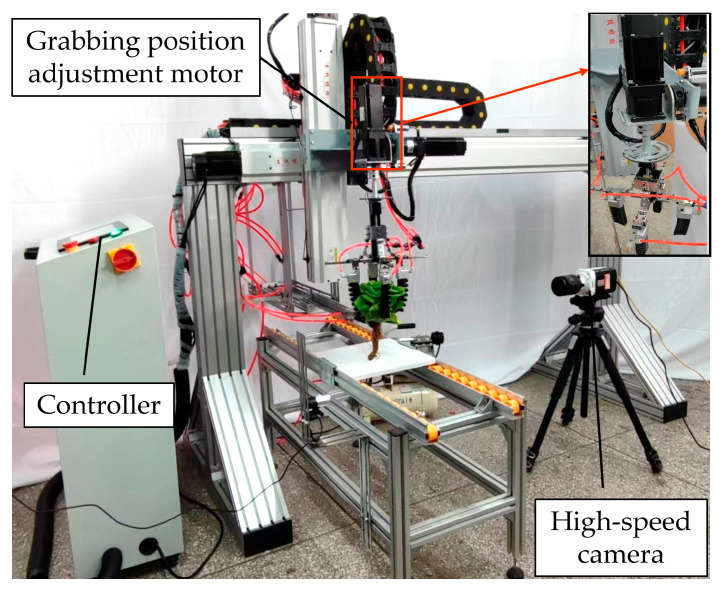
High-speed photography acquisition system.

**Figure 6 sensors-23-06047-f006:**
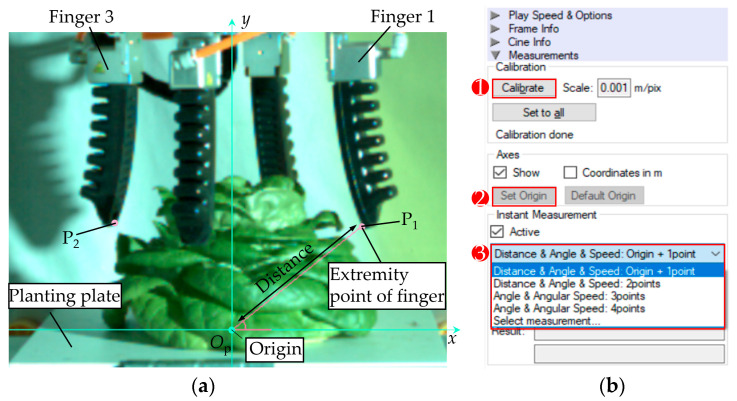
Measurement of flexible finger motion characteristics using high-speed photogrammetry: (**a**) measurement point selection and coordinate system; (**b**) software operation procedure.

**Figure 7 sensors-23-06047-f007:**
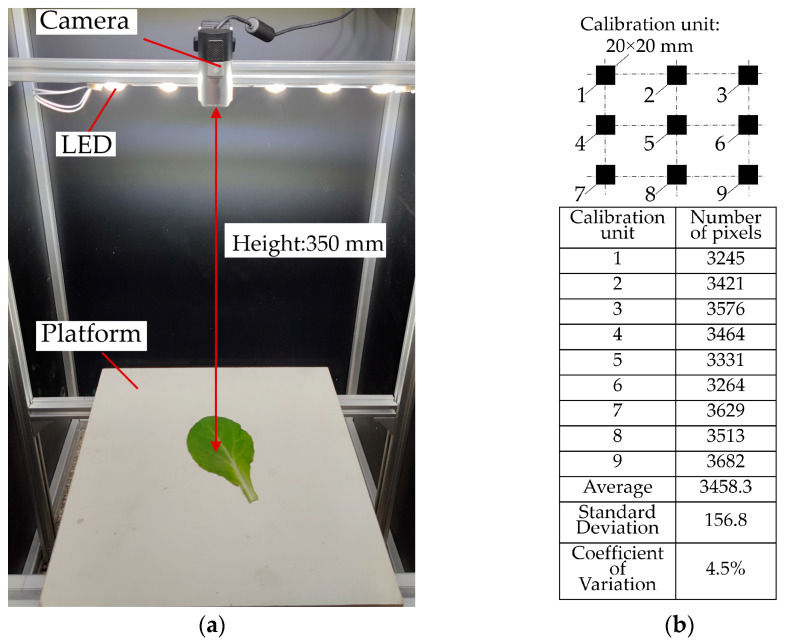
Image acquisition device and area calibration: (**a**) image acquisition device; (**b**) area calibration result.

**Figure 8 sensors-23-06047-f008:**
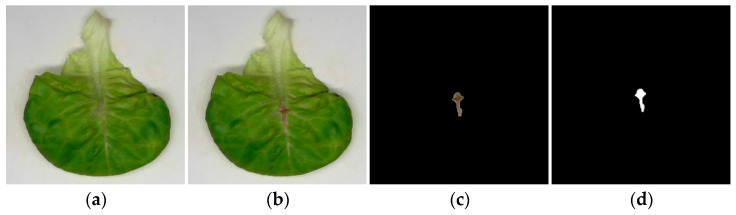
Measurement of injury areas of a vegetable leaf: (**a**) raw image of vegetable leaves; (**b**) stained image; (**c**) manual extraction of the injured area; and (**d**) calculation of injury area.

**Figure 9 sensors-23-06047-f009:**
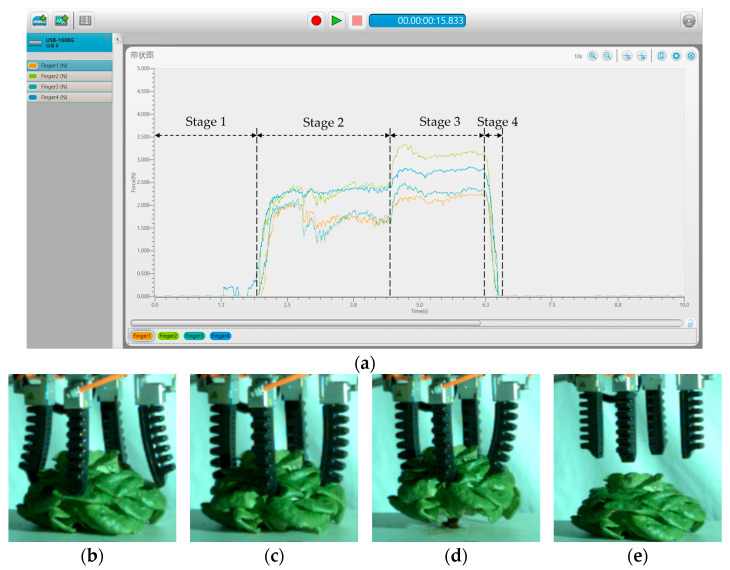
An example of a grabbing process (Grabbing Type B): (**a**) grabbing force measurement software interface; (**b**) Stage 1: fingers expanding and closing on lettuce; (**c**) Stage 2: fingers shrinking to grab lettuce; (**d**) Stage 3: lifting lettuce up; (**e**) Stage 4: releasing lettuce.

**Figure 10 sensors-23-06047-f010:**
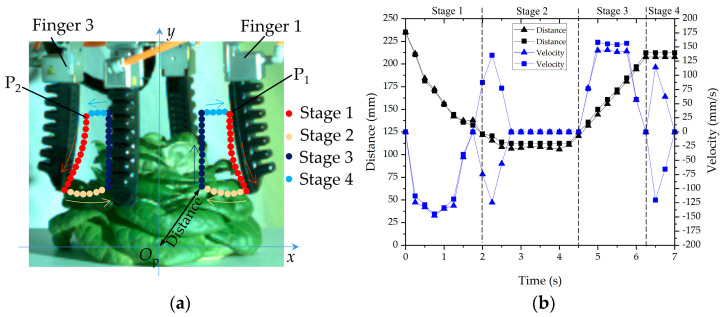
High-speed photographic analysis of contact points between flexible fingers and leaves: (**a**) trajectory of contact points; (**b**) motion characteristics of flexible contact points.

**Figure 11 sensors-23-06047-f011:**
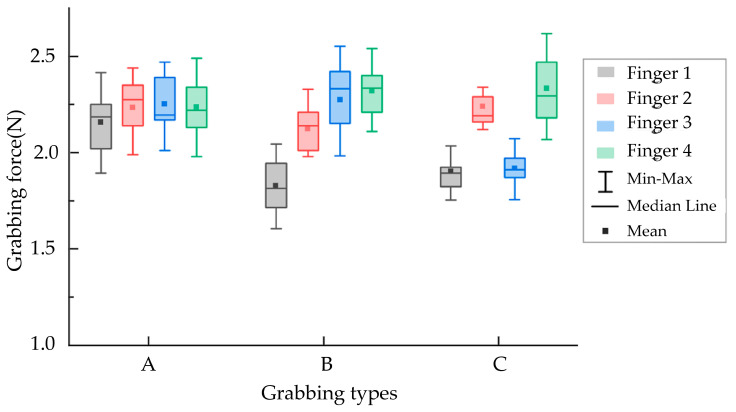
Grabbing force of each flexible finger under different grabbing types.

**Figure 12 sensors-23-06047-f012:**
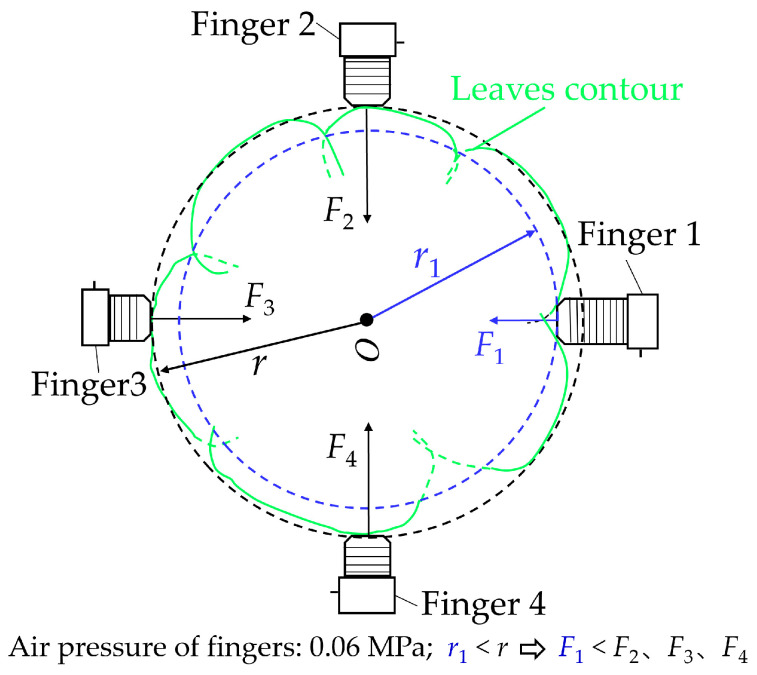
Grabbing force analysis of flexible fingers.

**Table 1 sensors-23-06047-t001:** Statistical analysis of force characteristics of four flexible fingers.

Stage	Finger	Maximum Force (N)	Mean Force (N)	Standard Deviation (N)	Coefficient of Variation (%)
1	1	0.06	0.01	0.007	70.0
	2	0.04	0.02	0.007	35.0
	3	0.05	0.02	0.007	35.0
	4	0.05	0.02	0.009	45.0
2	1	1.72	1.54	0.539	43.9
	2	2.45	2.08	0.631	41.4
	3	1.82	1.53	0.532	55.2
	4	2.37	2.10	0.539	39.6
3	1	2.15	2.12	0.107	50.4
	2	3.32	3.11	0.129	41.4
	3	2.47	2.28	0.136	59.6
	4	2.79	2.73	0.083	30.4
4	1	2.23	1.05	0.931	88.6
	2	3.12	1.62	1.109	67.9
	3	2.32	1.06	0.880	83.0
	4	2.74	1.60	0.926	57.8

**Table 2 sensors-23-06047-t002:** The accelerations of contact points at each grabbing stage.

Stage	Time (s)	P_1_ (mm/s^2^)	Direction	P_2_ (mm/s^2^)	Direction
1	0–0.75	69	y−	69	y−
	0.75–2	60	y+	60	y+
2	2–2.25	410	x−	445	x+
	2.25–2.75	215	x+	220	x−
	2.75–4.5	0	-	0	-
3	4.5–5.0	292	y+	310	y+
	5.0–5.75	0	-	0	-
	5.75–6.25	276	y−	289	y−
4	6.25–6.5	215	x+	220	x−
	6.5–7	106	x−	112	x+

**Table 3 sensors-23-06047-t003:** Average grabbing force of flexible fingers under various grabbing types.

	Finger	1	2	3	4	Standard Deviation (N)	Coefficient of Variation (%)
Grabbing Type							
Mean force of Type A		2.16 N	2.23 N	2.25 N	2.23 N	0.042 N	1.9%
Mean force of Type B		1.67 N	2.12 N	2.30 N	2.32 N	0.302 N	14.4%
Mean force of Type C		1.79 N	2.24 N	1.83 N	2.33 N	0.278 N	13.6%
Standard deviation		0.26 N	0.07 N	0.26 N	0.06 N	/	/
Coefficient of variation		13.6%	3%	12.1%	2.4%	/	/

**Table 4 sensors-23-06047-t004:** Injury area of the leaves under different grabbing types.

	Grabbing Type	A	B	C
Leaves Injury Area (mm^2^)				
1		284	195	141
2		255	219	104
3		260	187	114
4		292	196	125
5		296	199	115
6		282	229	137
7		250	184	135
8		287	224	117
9		269	214	102
10		280	221	113
Average value (mm^2^)		275.5	206.8	120.3
Standard deviation (mm^2^)		16.0	16.4	13.7
Coefficient of variation (%)		5.8	7.9	11.4

## Data Availability

Not applicable.

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
