# Peer review of "Effects of Harvesting Grabbing Type on Grabbing Force and Leaf Injury of Lettuce"

_sensors, 2023, doi:10.3390/s23136047_

Round 1

Reviewer 1 Report

This article describes Effects of grabbing position on grabbing force and leaves injury area in flexible harvesting of hydroponic lettuce in great detail. But there are still the following problems:

l  First of all, the writing of the thesis is not refined, and the language is lengthy, which reflects that the author's ability to refine problems and generalize needs to be improved. In particular, the abstract and introduction part need to be revised. The analysis of the current research status at home and abroad has not been refined and summarized. The keywords do not correspond well to the content of this article, and still need to be modified. For example, "plant factory" is only one of the application backgrounds of the research object of this article, and it is not suitable for the content of this article. The summary of the current status of related research in the introduction is more like a simple accumulation list, and no clear logic can be seen. References should be more cutting edge.

l  The pictures in the full text are not clear, and the layout and font settings are unreasonable, such as "the color of the label in Figure 2 is unreasonable, the overlap is unclear, and there are problems with the line type and line thickness, and the expression of d2 and bending angle θ is very unclear" . The picture in Figure 9 needs to be modified, the picture is unreasonable, please use the language specified by the journal.

l  In lines 110-113, what are the reasons for the three crawl types defined? It is not clear why there are only these three crawling situations. Due to the different postures and growth conditions of the lettuce plants, one finger grasps the leaf surface, three fingers grasp the leaf cross, or all four fingers grasp the leaf cross, and there is a lack of discussion and description of these situations. In addition, should the impact of the grab height position also need to be considered? In the case of different heights, the radial positions of the leaf surface and leaf exchange are also different.

l  In lines 128-130, the experimental protocol employs four thin-film pressure sensors attached to the bottom of the flexible finger. However, the pressure sensor is a small circular shape, which cannot cover all the grasping contact area, and the flexible finger grasping type does not only work on the target at the bottom. The above circumstances will cause obvious errors, is the test result reliable, and is the test plan reasonable?

l  In lines 156-161, there are only two measurement points selected for the test, and the purpose of the test is to analyze the motion characteristics of the flexible fingers during the grasping process. There are obvious uncertainties in the gripping process, just like the different gripping types mentioned above, how can it be ensured that the two measurement points represent all situations? The test plan should be improved, set measurement points for all four fingers, perform multiple grasps, clarify the measurement point data corresponding to each grasp type, and then analyze the motion characteristics of flexible fingers.

l  The position of the damaged area shown in Figure 9 is not consistent with the grasping position. What is the reason? Is there any difference in leaf damage in different layers? Analysis can be carried out in a targeted manner.

Minor editing of English language required

Author Response

Distinguished Reviewer,

Thanks for your reviews. We appreciate your suggestions raised to improve the manuscript. We have revised the manuscript accordingly and marked up the revisions using the “Track Changes” function. The responses to the comments were explained point by point, the revised content and important information was marked with underlines, Please see the attachment (Response to Reviewer 1 Comments.doc).

Reviewer 2 Report

The authors present an  interesting manuscript on the Effects of grabbing position on grabbing force and leaves injury area in flexibe harvesting of hydroponic lettuce. The following are my specific comments:

1. The topic could be revised to read ' Effects of harvesting grabbing type on grabbing force and leaf injury of lettuce'.

2. The objectives should be rephrased especially the first objective. What do the authors mean by ' to study the law of leaf arrangement...."?

3. Authors are encouraged to merge the Experimental material and exquipment and Expeirmental methods Under a broad heading Materials and Methods. The earlier topics can be sub-titles under the Materials and Methods.

4. The manuscript should have a section for Data or Statitistical analysis. Also, I did not see the experimental design used in this work.

5. Under the Results and Discussion, the authors only presented the results of the findings without discussing it in the context of current literature. This is unacceptable.

6. The conclusion should be rephrased especially for the first one.  Conclusion should answer the stated objectives.

The authors should get a native speaker to edit the language for easy understanding by readers.

Author Response

Distinguished Reviewer,

Thanks for your reviews. We appreciate your suggestions raised to improve the manuscript. We have revised the manuscript accordingly and marked up the revisions using the “Track Changes” function. The responses to the comments were explained point by point, the revised content and important information was marked with underlines, Please see the attachment (Response to Reviewer 2 Comments.doc).

Reviewer 3 Report

The grasping impact of 4-finger flexible fingers on the hydroponic lettuce at different positions were studied, which provided a reference for the development of hydroponic lettuce harvesting equipment.  The following problems are suggested to be further modified:

(1) In the introduction, the research background of flexible grabbing method utilized of this study was not clearly expressed.  Why the grabbing method with four flexible finger is worth further study?  Why was the grabbing position considered to be an important influence factor for leave injury by the impacts of flexible finger?

(2) In the “3.  Experimental methods”, how to prove the injury detection method is reasonable and effective?

(3) How to utilize the research results of the study to solve practical harvest problem?  It should be further discussed.

Minor editing of English language.

Author Response

Distinguished Reviewer,

Thanks for your reviews. We appreciate your suggestions raised to improve the manuscript. We have revised the manuscript accordingly and marked up the revisions using the “Track Changes” function. The responses to the comments were explained point by point, the revised content and important information was marked with underlines. Please see the attachment (Response to Reviewer 3 Comments.doc).

Round 2

Reviewer 1 Report

It is ok.

It is ok.

Reviewer 2 Report

Authors have responded to most of my earlier comments well. However, an Enlgish Editor or speaker should help to fine tune the grammar in the mansucript to help readers understand the content of the work.

The authors should get an English native speaker and writer to fine tune the grammar in the work.